# Genome-Wide Identification, Diversification, and Expression Analysis of Lectin Receptor-Like Kinase (LecRLK) Gene Family in Cucumber under Biotic Stress

**DOI:** 10.3390/ijms22126585

**Published:** 2021-06-19

**Authors:** Muhammad Salman Haider, Savitha De Britto, Geetha Nagaraj, Bhavya Gurulingaiah, Ravikant Shekhar, Shin-ichi Ito, Sudisha Jogaiah

**Affiliations:** 1Key Laboratory of Genetics and Fruit Development, College of Horticulture, Nanjing Agricultural University, Nanjing 210095, China; salman.hort1@gmail.com; 2Laboratory of Plant Healthcare and Diagnostics, P.G. Department of Biotechnology and Microbiology, Karnatak University, Dharwad 580003, India; savitha.debritto51@gmail.com; 3Division of Biological Sciences, School of Science and Technology, University of Goroka, Goroka 441, Papua New Guinea; 4Nanobiotechnology Laboratory, Department of Studies in Biotechnology, University of Mysore, Manasagangotri, Mysuru 570006, India; geethabiotech.uom@gmail.com (G.N.); bhavyagurulingaiah@gmail.com (B.G.); ravikantshekhar@gmail.com (R.S.); 5Laboratory of Molecular Plant Pathology, Department of Biological and Environmental Sciences, Graduate School of Sciences and Technology for Innovation, Yamaguchi University, Yamaguchi 753-8515, Japan; 6Research Center for Thermotolerant Microbial Resources (RCTMR), Yamaguchi University, Yamaguchi 753-8515, Japan

**Keywords:** cucumber, lectin receptor-like kinases, plant growth-promoting fungi, biotic stress

## Abstract

Members of the lectin receptor-like kinase (LecRLKs) family play a vital role in innate plant immunity. Few members of the LecRLKs family have been characterized in rice and *Arabidopsis*, respectively. However, little literature is available about LecRLKs and their role against fungal infection in cucumber. In this study, 60 putative cucumber LecRLK (CsLecRLK) proteins were identified using genome-wide analysis and further characterized into L-type LecRLKs (24) and G-type LecRLKs (36) based on domain composition and phylogenetic analysis. These proteins were allocated to seven cucumber chromosomes and found to be involved in the expansion of the *CsLecRLK* gene family. Subcellular localization of CsaLecRLK9 and CsaLecRLK12 showed green fluorescence signals in the plasma membrane of leaves. The transcriptional profiling of *CsLecRLK* genes showed that L-type LecRLKs exhibited functional redundancy as compared to G-type LecRLKs. The qRT-PCR results indicated that both L- and G-type LecRLKs showed significant response against plant growth-promoting fungi (PGPF-*Trichoderma harzianum* Rifai), powdery mildew pathogen (PPM—*Golovinomyces orontii* (Castagne) V.P. Heluta), and combined (PGPF+PPM) treatments. The findings of this study contribute to a better understanding of the role of cucumber *CsLecRLK* genes in response to PGPF, PPM, and PGPF+PPM treatments and lay the basis for the characterization of this important functional gene family.

## 1. Introduction

Powdery mildew of cucumber (*Cucumis sativus* L.) caused by *Golovinomyces orontii* (Castagne) V.P. Heluta. is a devastating fungal disease across the world that causes major damage to most of the Cucurbitaceae plants, which affects the quality of the fruit, thereby resulting in high yield loss in both greenhouse and field crops [1]. The management of powdery mildew disease is a challenging task in the cucumber growing regions. Systemic fungicide, flusilazole, and potassium salts as inducers are effectively recommend to overcome or reduce the disease problem. However, the application of such chemicals can result in severe negative impacts on human health and environmental contamination. Hence, eco-friendly disease management practices are essential to overcome with this disease. Amongst these eco-friendly approaches, plant growth promoting fungi (PGPF) isolated from plant rhizosphere that stimulates host immunity, resistance, or protection [2] by triggering lectin receptor-like kinases (LecRLKs) against biotic stress has gained considerable importance for the development of sustainable crop improvement [3,4].

Cell to cell surface interactions through receptors often controls growth and development in multicellular organisms. In general, the receptors of the cell surface play a key role in recognizing and establishing cellular signal events [4]. Receptors like kinases (RLKs) are one such group of surface receptors, and are the largest group of proteins, consisting of more than 1% of protein encoding genes in *Arabidopsis* [5]. There are two groups of RLKs that play an important part in extracellular and intracellular signaling, having extremely divergent proteins with an LRR domain, a lectin domain, and a kinase domain as major conserved regions. Among RLKs, the leucine-rich repeat receptor like kinase (LRR-RLK) family is identified as the largest family and consists of about 185 putative LecRLKs genes in the soybean genome [6]. Based on the phylogenetic and structure analysis of the kinase domains, these RLK proteins of *Arabidopsis thaliana* (L.) Heynh. were grouped into 15 families [5]. Due to greater expansion of land, the linkage of plants has caused the LRR-RLK gene family to emerge as a means of plant adaptation, particularly for sensing environmental signals [7]. The LRR -RLKs possess extracellular carbohydrate features that are capable of binding lectin domain and play a key role in developmental stages of a plant, such as morphogenesis, organogenesis, hormone signaling, and innate immunity such as response to environmental cues like stress and defense related actions. Thus far, 15 sub families of RLKs have been recognized, based on the extracellular domains [5]. Lectin receptor-like kinases (Lec RLKs) are vital factors in response to plant defense against bio-trophic pathogens. To date, 93 tomato sly *LecRLKs* genes involving 23 L type, 69 G type, and one C type *Lec RLK* have been identified using genome analysis [8]. Lec RLKs are lectin domains that have similar features to lectin proteins that bind to carbohydrates found in animals and humans [9,10]. Membrane bound Lec RLKs are important in the cellular responses to various biotic, abiotic stress, as well as developmental signs [4]. It is also documented that leucine rich repeat receptors like kinase (LRR-RLKs) controls growth and development against various stresses. For instance, LRR-RLKs in *Arabidopsis* are BRI1 (Brassinosteroid Insensitive 1) and CLV1 (Clavata 1) [11,12]. CLV1 is known to balance the specification of stem cells in flower meristems and aid in the detection of CLVs domains. Likewise, a steroid hormone BRI1 is the receptor for brassinolide (BL), which is involved in stimulation of seed size, elongation of stem, differentiation of vascular bundles, fertility, senescence, and flowering period [13,14]. Moreover, an *LRR-RLK* putative tomato gene in the genome helps to determine the phylogeny, gene organization, conserved motif, and expression of transcriptional events, as has been well investigated [14]. A total of 19 *SlLRR-RLK* genes, which have been designated *SlLRR-RLK1–SlLRR-RLK19*, express the presence of the five most highly expressed genes in 10 types of Heinz1706 organs or tissues. These have been further analyzed for expression levels [14] using transcriptome data [15].

Several L-type LecRLKs (e.g., *LecRK*-*IX*.*2* and *LecRK-1.9*) have been found to trigger pathogen triggered immunity (PTI) against *Pseudomonas syringe* infection [3]. Likewise, LecRK-V1.2 serves as a positive regulator of PTI in plants and activates a defense response against biotrophic and necrotrophic pathogens [16]. Moreover, G-type LecRLKs have been found to induce *Laccaria bicolor* symbiosis with plants. For example, overexpression of populous PtLecRLK1 in *A. thaliana* (a non-host for *L. bicolor*) showed symbiosis between *L. bicolor* and *A. thaliana* [17]. Lv et al. [18] quantified the expression of different LecRLKs in response to different hormonal and stress treatments and found that each gene has a distinctive expression pattern. To date, there are reports of about 213, 309, and 382 *LRR-RLK* genes that have been detected in rice, *Arabidopsis*, and poplar whole genome sequences, respectively [5], but no previous studies on the action of cucumber LecRLK against fungal infection have been reported. Hence, the present study is focused on identifying the presence and expression pattern of LecRLK genes that may trigger the cucumber plant’s innate defense system against biotic stress. Its effect against the devastating powdery mildew disease, in particular, is thoroughly investigated.

## 2. Results

### 2.1. Identification and Physicochemical Properties of CsaLecRLK Genes

After mining the orthologous genes of *Arabidopsis*, 60 putative CsLecRLK protein sequences were identified in total, which were further classified into L-type *LecRLKs* (*n* = 24) and G-type *LecRLKs* (*n* = 36) genes (Appendix A). These are denominated as *CsaLecRLK1–CsaLecRLK24* for L-type genes and *CsaLecRLK1–CsaLecRLK36* for G-type genes, according to the previously listed nomenclature, with the occurrence of an extracellular bulb lectin in each sequence. A summary including a brief description, such as chromosomal location, coding sequence (CDS) length (bp), and other protein properties (i.e., amino acid (AA) length, molecular weight (MW; kDa), isoelectric point (pI), grand average of hydropathocity (GRAVY), and subcellular prediction) for both L-types and G-types are shown in Appendix A. The results suggested that the CDS length ranged from 1518–2691 bp and 1923–2922 bp, whereas the protein lengths varied from 505–896 aa and 640–973 aa for L-type and G-type proteins, respectively. Moreover, the MW varied from 56.33–97.66 kDa for L-type and 71.82–108.31 kDa for G-type proteins, while pI ranged from 5.29–8.11 for L-type and 5.26–8.7 for G-type proteins. Similarly, the GRAVY analysis revealed that both L-type and G-type proteins are hydrophobic with negative values, except for the L-type gene *CsaLecRLK16* (0.042) and the G-type genes *CsaLecRLK5* and *CsaLecRLK6* (0.046 and 0.143), which intimated a positive hydrophilic nature. The subcellular prediction analysis asserted that both *L-type* and *G-type* genes reside in different cellular compartments, including plastid, vacuole, chloroplast, mitochondria, cytoplasm, nucleus, etc. (Appendix A). Finally, gene duplication analysis demonstrated a distinction between the duplication type for both L-type (4 tandem, 3 dispersed, 2 segmental, and 2 proximal) and G-type (8 segmental, 6 tandem, 1 dispersed, and 1 proximal) (Table 1). Taken together, the distinctions between L-type and G-type within genes intimate the diverse role of these genes in a variable environment.

### 2.2. Phylogenetic Relationship, Motif Composition and Gene Structure Analysis of CsaLecRLKs Genes

A phylogenetic tree of 60 putative *CsLecRLK* genes was constructed to validate the domain-based classification of L-type *LecRLKs* (Figure 1A, Appendix A) and G-type *LecRLKs* (Figure 1B, Appendix A) by MEGA (ver. 7.0) using maximum likelihood (ML). The tree consisted of 24 L-type *LecRLK* (40%) genes and 36 G-type *LecRLKs* (60%) genes, while no gene was classified as a C-type *LecRLKs* from the cucumber genome when compared with *Arabidopsis*. These findings are in line with the structural classification of the *LecRLK* gene family in other crops [8,19]. Moreover, a motif analysis of LecRLK amino acid sequences was carried out using the MEME program. The MEME analysis identified ten conserved motifs with diverse architecture for both L-type and G-type LecRLK proteins, respectively (Figure 2A,B). The results intimated that motif one to five predominantly occurred in L-type *LecRLK* genes (Figure 2A), whereas motif one, two, six, and eight were frequently found in G-type *LecRLK* genes (Figure 2B). Therefore, the MEME findings suggested that both L-type and G-type LecRLK proteins contain distinct features based on the variations in their amino acid sequences.

In addition, the gene structure of CDS and untranslated regions (UTRs) of G-type and L-type *LecRLK* genes in cucumber were illustrated using TBTools (Appendix A), which depicted a high level of divergence between G-type and L-type LecRLK members that were largely conserved. Interestingly, G-type and L-type *LecRLK* genes exhibited similarities between the same clades.

### 2.3. Chromosomal Location and Gene Duplication Analysis of CsaLecRLK Genes

The results of chromosomal locations of both L-type and G-type *LecRLK* genes showed variation in their positions and composition at different chromosomal sites. In L-type LecRLKs, the greatest number of genes were found at Chr2 (8), followed by Chr7 (6), and lowest numbers were present at Chr6 (1) (Figure 3A). Similarly, G-type *LecRLK* genes were unevenly distributed, indicating the highest number of genes at Chr1 (15), followed by Ch3 and Chr6 (6), while the lowest numbers were found at Chr2 (1) (Figure 3B). Moreover, both G-type and L-type *LecRLK* genes were clustered for collinear relation between *C. sativus* and *A. thaliana*, and within *C. sativus* (Figure 3A,B). The collinearity analysis signified higher conservation among L-type LecRLK proteins as compared to G-type LecRLK proteins (Appendix A).

The values of non-synonymous (*Ka*) and synonymous *(Ks)* substitution rates were determined to evaluate the selection pressure between duplication types of L-type and G-type LecRLKs. During pre-evolutionary history, many genes underwent various selection processes, including neutral, positive, and purifying selections. To comprehend the selective pressure, eleven L-type *LecRLK* and sixteen G-type *LecRLK* gene pairs were selected, and the results showed that the *Ka/Ks* ratio was less than one for most of the L-type and G-type *LecRLKs*, suggesting the purifying selection of these genes with less divergence after duplication (Table 1). Moreover, three gene pairs of each L-type *LecRLKs* (i.e., *CsaLecRLK2-CsaLecRLK4*, *CsaLecRLK6-CsaLecRLK7*, and *CsaLecRLK18-CsaLecRLK19*) and G-type *LecRLKs* (i.e., *CsaLecRLK2-CsaLecRLK3*, *CsaLecRLK28-CsaLecRLK29*, and *CsaLecRLK5-CsaLecRLK14*) showed positive selection. Additionally, we calculated the rate of divergence and the result suggested that the estimated divergence time of L-type *LecRLK* and G-type *LecRLK* gene pairs is 6.00-57.67 million years ago (MYA) and 7.33-45.33 MYA, respectively, which is much earlier than the emergence of *Arabidopsis.*

### 2.4. Gene Ontology (GO) and Promoter Analysis of CsaLecRLK Genes

For the functional annotation of G-type LecRLKs (*n* = 36) and L-type LecRLKs (*n* = 24) genes, GO-based enrichment analysis was performed. GO analysis functionally characterizes transcripts into three major terms, such as molecular function (MF), biological process (BP), and cellular component (CC) [20]. GO classified L-type *LecRLK* transcripts into four MFs, (e.g., ‘kinase activity’, ‘signaling receptor activity’, ‘transferase activity’, and ‘catalytic activity’), two CCs (e.g., ‘plasma membrane’ and ‘membrane’), and nine BPs (e.g., ‘response to biotic stress’, ‘cell communication’, ‘response to ‘cellular protein’, external stimulus’, ‘signal transduction’, ‘modification process’, ‘protein metabolic process’, ‘cellular process’, and ‘metabolic process’) (Figure 4A; Appendix A). Similarly, G-type *LecRLKs* were also categorized into four MFs (e.g., ‘nucleotide binding’, ‘kinase activity’, ‘transferase activity’, and ‘binding’), one CC (‘membrane’), and six BPs (i.e., ‘pollination’, ‘cell communication’, ‘reproduction’, ‘protein metabolic process’, ‘cellular protein modification process’, and ‘cellular process’) (Figure 4B; Appendix A). GO results for both L-type and G-type *LecRLK* genes suggested their critical development- and stress-related roles in plants.

The cis acting components were determined for the promoter regions of *CsaLecRLK* genes using the PlantCare database. The results showed that the majority of genes were involved in various signaling-related pathways such as light regulation (29.73%), followed by hormone signaling (26.61%), while less participation was found in other essential elements (11.6%) (Figure 5; Appendix A). Moreover, *CsaLecRLK* genes also exhibited participation in stresses (heat, drought, low temperature, etc.) and other regulatory stress factors (Box-W1, ELI-Box3, CE3, EIRE, etc.), deducing that these genes have multiverse roles and may function against various biotic/abiotic stress factors.

### 2.5. Subcellular Localization of CsaLecRLKs

The plasma membrane-based localization of CsaLecRLK proteins plays a critical role in cell wall and membrane links, as well as transmembrane movements, which govern the plant’s response to pathogen attack. For the subcellular localization, we transformed CsaLecRLK9 and CsaLecRLK12 with 35S-GFP into tobacco leaves. The results showed fluorescent green signals in the plasma membrane, suggesting that both (CsaLecRLK9 and CsaLecRLK12) proteins function from the plasma membrane (Figure 6).

### 2.6. Expression Pattern and qRT-PCR Validation of CsaLecRLK Genes

We mined RNA-sequencing data to provide insights into potential gene functions by analyzing the RPKM-based expression of *CsaLecRLK* genes present in various organs and tissues of the cucumber. The results indicated that numerous L-type LecLRKs (i.e., *CsaLecRLK2*, *CsaLecRLK9*, *CsaLecRLK12*, *CsaLecRLK13*, *CsaLecRLK15*, *CsaLecRLK16*, *CsaLecRLK20*, and *CsaLecRLK23*) showed higher expression in all organ or tissues, whereas *CsaLecRLK8*, *CsaLecRLK15*, *CsaLecRLK17*, and *CsaLecRLK19* demonstrated weak expression in all selected tissues and organs (Appendix A). Among G-type LecRLKs, *CsaLecRLK8*, *CsaLecRLK24*, *CsaLecRLK26*, *CsaLecRLK29*, *CsaLecRLK31*, and *CsaLecRLK33* revealed organ- or tissue-specific responses in all tissues, whereas the others showed moderate to low expression in the observed cucumber tissues (Appendix A). Taken together, these findings suggest that L-type *LecRLK* genes have more profound organ- or tissue-specific responses than G-type *LecRLKs*. Moreover, previously published RNA-sequence data on various biotic (*Botrytis cinerea* Pers.) and abiotic (drought, alkali, and nitrogen deficiency) stressors were obtained to compare the *LecRLK* gene family response at a post-transcriptional level. The findings revealed that both (L-type and G-type) groups showed low to moderate expression in response to drought and alkali stress; however, they exhibited a moderate to high response towards N-deficiency and *B. cinerea* infection. Comparatively, the response was more profound in cucumber leaves responding to a fungal infection caused by *B. cinerea* (Figure 7A,B).

To comprehend the critical role of LecRLKs in abiotic stress resistance, the qRT-PCR validation of sixteen randomly selected genes of both L-type *LecRLKs* and G-type *LecRLKs* was carried out in three different treatments, including plant growth-promoting fungi (PGPF), *Golovinomyces orontii* pathogen (PPM), and a combination of the two (PGPF+PPM). The results demonstrated a distinction in the expression patterns of both L-type and G-type LecRLK genes (Figure 8A,B). Out of 16 L-type *LecRLKs*, six showed up-regulation against PGPF, five against PPM, and four against the combination (PGPF+PPM) treatment. During PGFP treatment, several genes, including *CsaLecRLK1*, *2*, *3*, and *13* showed significantly higher up-regulation when compared with the control. Moreover, *CsaLecRLK6*, *10*, and *16* exposed to PPM and *CsaLecRLK10*, *12*, and *13* exposed to PPM+PGPF suggested significant up-regulation. Among 16 G-type LecRLK genes, nine *LecRLKs* (i.e., *CsaLecRLK1*, *5*, *9*, *10*, *11*, *13*, *14*, *15*, and *16*) exposed to PGPF, seven *LecRLKs* (*CsaLecRLK1*, *3*, *4*, *5*, *7*, *9*, and *10*) exposed to PPM, and three *LecRLKs* (*CsaLecRLK1*, *6*, *10*, and *12*) exposed to PGPF+PPM intimated up-regulation as compared to their control. Intriguingly, *CsaLecRLK1* and *CsaLecRLK10* of both (L-type and G-type) groups were commonly expressed in all treatments, signifying their role in combatting fungal and bacterial infections. Overall, qRT-PCR findings suggested that both L-type and G-type *LecRLK* genes showed vital defense-related responses in cucumber against biotic stresses.

## 3. Discussion

The LecRLKs are membrane-bound protein kinases that are known to play a vital role during growth and development, environmental stress responses, and pathogen attacks [6,8]. The genome-wide investigation into LecRLKs has been thoroughly studied in various crops, such as soybean [16] and tomato [8]. Previous studies on genome-wide analysis have identified *LecRLK* genes in many crops, including *Arabidopsis* (75), wheat (263), and rice (173) [4,6,21], which signifies that differences in copy numbers of LecRLKs might be due to variation in genome size and expansion rate. However, this family has not yet been studied extensively in cucumber plants. In this study, 60 putative *LecRLK* genes were detected from a cucumber genome and were further divided into the L-type *LecRLKs* (24) and G-type *LecRLKs* (36). We performed analyses of physicochemical properties, linkage mapping, chromosomal locations, collinearity, promoter, gene structure composition, and gene duplication. Furthermore, GO, cis-regulatory elements, and transcriptional dynamics between different organs of cucumber were also analyzed, which depicted extensive information and variation between G-type *LecRLK* and L-type *LecRLK* genes. The variation in physicochemical and protein properties (e.g., amino acid (AA) length, MW (kDa), pI, and GRAVY) suggested the occurrence of novel LecRLK variants in cucumber.

To comprehend the evolutionary relationship, two separate phylogenetic trees for both G-type and L-type *LecRLK* genes were constructed using the MEGA program (7.0) with the maximum likelihood (ML) method, since L-type and G-type LecRLKs have evolved separately. The results revealed varied bootstrap values for different nodes, which might be due to variation in the sequence among different clades [22]. In cucumber, G-type LecRLKs (*n* = 36) proteins are more numerous than L-type LecRLks (*n* = 24), which is different from *Arabidopsis* (L-type: *n* = 42 vs. G-type: *n* = 32). Hence, the number of LecRLK proteins in angiosperm is variable, which could be due to differences in selection pressures and the expansion rates of the genome. The expansion between G-type and L-type greatly varies and ranges from 0.085–0.323% for L-type and 0.117–0.449% for G-type, indicating that G-type LecRLKs expanded to a significantly greater scale as compared to L-type LecRLks. However, the expansion of the LecRLK gene family is highly uncoordinated in rice and *Arabidopsis* because orthologous gene pairs of this family in both species expanded at variable rates [4]. Moreover, estimated divergence time of L-type *LecRLK* genes and G-type *LecRLK* genes is 6.00–57.67 million years ago (MYA) and 7.33–45.33 MYA, respectively, which is much earlier than the *Arabidopsis* divergence time, and consistent with previous findings [23]. Our results also suggested that tandem duplications were more common in L-type *LecRLK* gene pairs, however, segmental duplication were highly observed in G-type *LecRLK* gene pairs, while previous reports revealed that both tandem and segmental duplications function in the amplification of *LecRLK* genes [24]. Furthermore, selection pressure analyses for both L-type and G-type *LecRLK* gene pairs suggested purifying selection (*Ka/Ks* < 1) for most of the genes, except very few that showed positive selection (*Ka/Ks* > 1). Henceforth, we conclude that LecRLK genes might duplicate prior to their existence and possess distinct functions.

The *cis*-acting elements within the promoters of genes are the transcriptional regulators of gene activities critically involved in hormone signaling, development and various stress responses [24,25]. In the current investigation, many biotic and abiotic stress responsive cis-regulatory elements that were identified, such as elicitor-responsive elements (jasmonates), are known to induce gene expression in bacterial mediated infection, while MeJA-responsive elements (CGTCA-motif and TGACG-motif) are very effective against disease infestation [26,27]. Moreover, DRE (dehydration responsive elements) are involved in drought, HSE (heat-shock responsive elements) against high-temperature stress, and LTR (low-temperature responsive elements) in chilling stress [28,29,30].

The tissue-specific expression abundance provides clues about gene biological functions [31]. Therefore, we mined RNA-sequence data that might offer insights into the potential tissue-specific functions of both L-type and G-type CsaLecRLK proteins (Figure 7A,B). The results of the transcriptional profiling reported the potential involvement of L-type and G-type *LecRLK* genes in organ development of the cucumber plant. Intriguingly, several genes of L-type *LecRLK* (e.g., *CsaLecRLK2*, *CsaLecRLK9*, *CsaLecRLK12*, *CsaLecRLK13*, *CsaLecRLK15*, *CsaLecRLK16*, *CsaLecRLK20*, and *CsaLecRLK23*) showed their regulation in all tissues, suggesting their functional divergence. However, very few of the G-type *LecRLK* genes (e.g., *CsaLecRLK8*, *CsaLecRLK24*, *CsaLecRLK26*, *CsaLecRLK29*, *CsaLecRLK31*, and *CsaLecRLK33*) showed tissue-specific expression in all, or even many, of the tissues, signifying their functional redundancy.

It has been demonstrated that G-type LecRLKs play a role against drought, salt, and ABA stresses [32], and induce resistance against *M. grisea* infection [33]. Based on previously published RNA-sequence data, the *CsaLecRLK* gene family showed low to moderate expression against drought and alkali, and moderate to high expression in response to N-deficiency and fungal infection. Ma et al. [34] investigated the role of LecRLKs against salinity stress in pears and revealed that six selected LecRLKs were shown to be salt-responsive. To verify the *CsaLecRLKs* role in pathogen infection, qRT-PCR validation was carried out in three different treatments (PGPF, PPM, and PGPF+PPM). The findings indicated functional diversity within L-type and G-type LecRLKs against all treatments. The expression pattern of some cucumber LecRLKs showed low expression or repression, perhaps because these genes are least affected by the pathogen infection. The expression of tobacco NtlecRK1 in tobacco bright yellow cells (BY-2) was attenuated upon elicitin (INF1) and bacterial elicitor harpin [35], which is consistent with the up-regulation *CsaLecRLK1* in all the treatments. Another study depicted mutants lacking *Arabidopsis LecRK1.9* showing enhanced susceptibility to fungal and bacterial infections [36]. The cucumber L-type *CsaLecRLK2* was significantly up-regulated in PGPF treatment, while a similar study demonstrated that cotton L-type LecRK-2, upon treatment with cell wall extracts, showed an elicitin-defense response [37]. In *Haynaldia villosa*, transgenic plants overexpressing L-type LecRLK (*LecRK*-*V*) were more affected by wheat powdery mildew [38]. Similarly, rice Pi-d2 (G-type LecRLK) has been found to induce the plant’s defense response against fungal pathogens [39], which is in agreement with our findings. These results broaden our understanding of the critical role of cucumber LecRLKs in plant-fungal interaction. However, more experiments are needed to functionally verify the L-type and G-type *LecRLKs* for host resistance against biotic stresses.

## 4. Materials and Methods

### 4.1. Identification and Sequence Retrieval for CsaLecRLK

The *cucurbitaceae* genome (http://www.cucurbitgenomics.org/ accessed on 17 April 2020) was searched to identify and retrieve the *LecRLK* sequences in cucumber, whereas the TAIR (http://www.arabidopsis.org/ accessed on 17 April 2020) database was mined to search for and retrieve the sequence for *Arabidopsis*. Retrieved protein sequences were then verified for the LecRLK domain using the SMART (http://smart.embl-heidelberg.de/ accessed on 18 April 2020) database [40]. The proteins lacking LecRLK domain and sequence lengths of less than 100 were removed from further analysis. (http://smart.embl-heidelberg.de/ accessed on 18 April 2020) [40]. Those proteins that lack a CsaLecRLK domain were removed from further analysis. In addition, protein sequences with obvious errors in their gene length, like being smaller than 100, were removed.

### 4.2. Phylogenetic Analysis of CsaLecRLK

The protein sequences of LecRLKs from both cucumber (Appendix A) and *Arabidopsis* were aligned by the multiple sequence alignment (MUSCLE) [41] and phylogenetic tree was constructed by maximum likelihood (ML) method with 1000 bootstrap values, following the Jones, Taylor, and Thornton (JTT) model by keeping other parameters default using MEGA software (V 7.0) [42].

### 4.3. Ka/Ks for Duplicated CsaLecRLK Genes and Their Rate of Divergence

The ratio of Ka/KS for duplicated gene pairs, such as tandem, proximal, segmental, and dispersed was calculated using the MEGA program (V 7.0) [42]. Moreover, the divergence time was calculated using the following formula: T = Ks/2r, in which r was taken to be 1.5 × 10^−8^ (synonymous substitution/year) by showing the rate of divergence [43].

### 4.4. Conserved Motifs, Exon-Intron Structure Analysis, and Physicochemical Parameters of CsaLecRLK Proteins

The identification of conserved motifs for CsaLecRLK proteins was accomplished using the local MEME Suite (V 5.0.3), and parameters were set to: maximum number of motifs 10, with a minimum width of 100 and a maximum of 150, while keeping the other parameters default [44]. Moreover, online Gene Structure Display Server (GSDS 2.0) (http://gsds.cbi.pku.edu.cn accessed on 21 April 2020) was used for exon and intron structure display [45]. The physicochemical properties, such as molecular weight (MW), isoelectronic points (pI), aliphatic index and GRAVY for each CsaLecRLK protein were calculated using the ExPASY PROTPARAM tools (http://web.expasy.org/protparam/ accessed on 21 April 2020). The subcellular prediction was carried out through the WOLF PSORT (https://wolfpsort.hgc.jp/ accessed on 25 April 2020) website.

### 4.5. Cis-Elements Predictions of CsaLecRLK

The Generic File Format (GFF) obtained from the cucumber genome was used to search the CsaLecRLK promoter sequence (selected as 1500 upstream bp). Afterwards, the online PlantCARE database (http://bioinformatics.psb.ugent.be/webtools/plantcare/html/ accessed on 7 May 2020) was employed to identify the cis acting elements of each protein [46].

### 4.6. Chromosomal Location and Syntenic Relationship, and Gene Onotology Enrichment Analysis

The location and genomic positions of each *CsaLecRLK* gene at their specific chromosome were illustrated on the circular diagram by using the TBTools program [47]. For syntenic relationships between the homologs of *A. thaliana* and *C. sativus* and within *C. sativus*, the Circos program was used and the synteny relationship was illustrated in different colors by using the TBTools program [47]. The GO enrichment test was performed using the online Panther Server (http://pantherdp.org/ accessed on 7 May 2020) and the figure was drawn using the TBTools program [48].

### 4.7. Subcellular Localization of CsaLecRLKs

To determine the CsaLecRLK protein localization, the full length CDSs of CsaLecRLK9 and CsaLecRLK12 were cloned and transiently transformed into the pCAMBIA1302 vector to produce CsaLecRLK9-GFP and CsaLecRLK12-GFP, and then transformed into agrobacterium (strain EHA105). The CDS was amplified using the PCR and inserted into the 35S-GFP (positive vector). The images were taken using an LSM 510 microscope (Zeiss Microscopy GmbH, Jena, Germany) with the 488nm laser line of an argon laser (50 mW). Images were taken under GFP, brightfield using a scale of 25 μm. All transient expression assays were repeated at least three times.

### 4.8. Plant Materials and Treatment

Cucumber local variety seeds (cv. Green long), that is ones that are highly susceptible to powdery mildew disease, were obtained from Eurogen Seeds Pvt. Ltd., Bangalore, Karnataka, India. The highly virulent cucumber powdery mildew pathogen, *G. orontii* was collected from Prof. Vijayendra B. Nargund, University of Agricultural Sciences, Dharwad, India. The pathogen was cultured on potato dextrose agar (PDA) medium and the seeds were kept at 4 °C for further experimental use.

### 4.9. Seed Priming with PGPF-Trichoderma Harzianum Isolate (TriH_JSB27)

Powdery mildew highly susceptible cucumber seeds were primed with 25 mL of conidial suspension of potent *T. harzianum* isolate (TriH_JSB27) (accession number, JQ665259) at a rate of 1 × 10^7^ spores/mL for 9 h under an orbital rotary shaker (70 rpm) (KEMI, KOSI-1, India) at room temperature (23 ± 2 °C) to uniform penetration of *T. harzianum* seeds. Seeds soaked with sterile distilled water (SDW) under the same conditions were maintained as controls. These treated and control seeds were then sown in earthen pots (13–15 cm diameter) filled with soil, sand, and farmyard manure (1:2:1) and grown in greenhouse conditions (4 plants per pot). One-month-old cucumber plants from primed PGPF, *T. harzianum* (four plants each) were challenge inoculated by spraying 100 mL of pathogen suspension, *G. orontii* (1 × 10^8^ spores/mL) till complete run-off. The control plants were sprayed with pathogen and or SDW alone and the experiment was performed twice.

### 4.10. RNA Isolation and Expression Profiling of C. sativus under Biotic Stress

Total RNA was isolated from the treated and control frozen leaves (day after pathogen or SDW inoculation) with Trizol (Invitrogen) following the manufacturer’s instructions. RNA was reverse-transcribed into cDNA using the Primer Script RT reagent kit (TAKARA, Dalian China) according to their instructions. Specific primers were designed using Becan Designer 7.9, and are presented in Appendix A. In order to check the specificity of the primers, we used the BLAST tool against the *C. sativus* genome for confirmation. RT-PCR was performed according to the guidelines of previous studies [49]. The relative expression level of CsaLecRLK genes was calculated against the actin as a reference gene for for qRT- PCR. The ABI 7500 Real Time PCR System (Applied Biosystems, Foster City, CA, USA) was used for amplification using the SYBR Green (Vazyme, Nanjing, China) with three biological replicates. The template for amplification were set as follows: denaturation at 95 °C for 10 min, 40 cycles of denaturation at 95 °C for 15 s, annealing at 60 °C for 15 s, and extension at 72 °C for 15 s.

The RNA-sequencing data on different tissues of cucumber against accession numbers, such as SRR351906 (cucumber leaf tissue), SRR351910 (cucumber tendril tissue), SRR351499 (cucumber root tissue), SRR351912 (cucumber female flower tissue), SRR351908 (cucumber male flower tissue), and SRR351905 (cucumber stem tissue), were obtained from the SRA database. Moreover, previously reported RNA-sequencing data on different biotic and abiotic stresses, including *Botrytis cinerea* (E-GEOD-72191), alkali (E-GEOD-42439), nitrogen deficiency (E-GEOD-46678), and drought (PRJNA219226) were obtained from the NCBI database. The FPKM (fragments per kilobase of transcript per million fragments mapped) values were used to quantify the transcription level, and a heatmap was generated on the basis of Log2 value using RStudio (R program) [50].

## 5. Conclusions

Previously, the role of *LecRLK* genes has been extensively studied against abiotic/biotic stress or hormonal response in many crops [42,43]. However, the potential function of *CsaLecRLK* genes against PGPF, PPM, and PGPF+PPM in the cucumber pathosystem has not yet been evidenced. The RNA-sequencing data was obtained to understand the tissue-specific response, and biotic and abiotic stress response of the *CsaLecRLKs*. Results suggest the functional variability of both L-type and G-type *CsaLecRLKs* in different cucumber tissues, and repose to different biotic and abiotic stresses. The expression analysis suggested that several genes from both L-type and G-type family showed up-regulation upon PGPF, PPM, and PGPF+PPM treatments, of which *CsaLecRLK1* and *10* were commonly expressed in both families in response to all treatments. Taken together, these results suggest that some LecRLK may function against the biotic stress tolerance in cucumber, which could help in a breeding program for developing durable powdery mildew resistant cucumber improved hybrids.

## Figures and Tables

**Figure 1 ijms-22-06585-f001:**
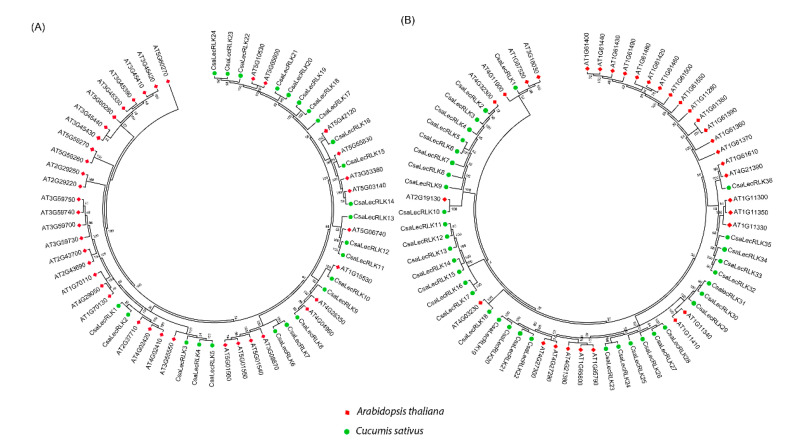
Phylogenetic analysis of (**A**) L-type LecRLK and (**B**) G-type LecRLK proteins between cucumber and *Arabidopsis*. The phylogenetic tree was constructed using MEGA (7.0) software with the maximum likelihood (ML) method with 1000 bootstrap values.

**Figure 2 ijms-22-06585-f002:**
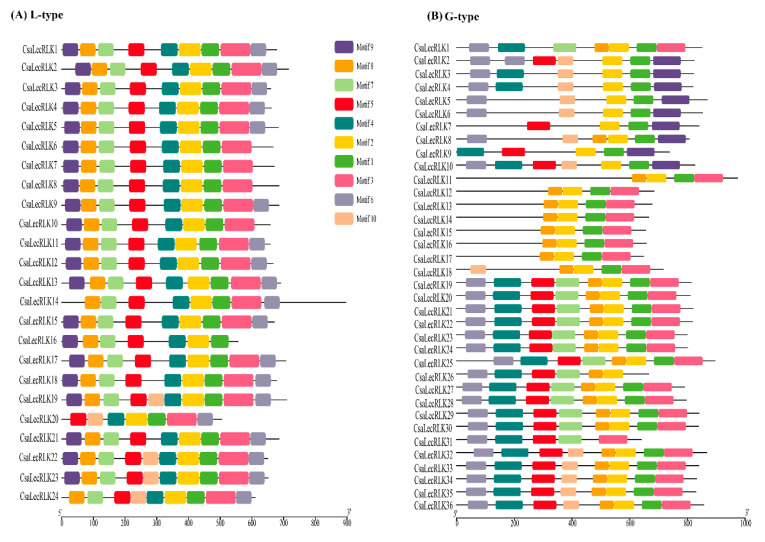
Conserved motif structure and up-/downstream regions of (**A**) L-type LecRLK and (**B**) G-type LecRLK proteins. The relative position is proportionally displayed based on the kolibase (KB) scale at the bottom of the figure.

**Figure 3 ijms-22-06585-f003:**
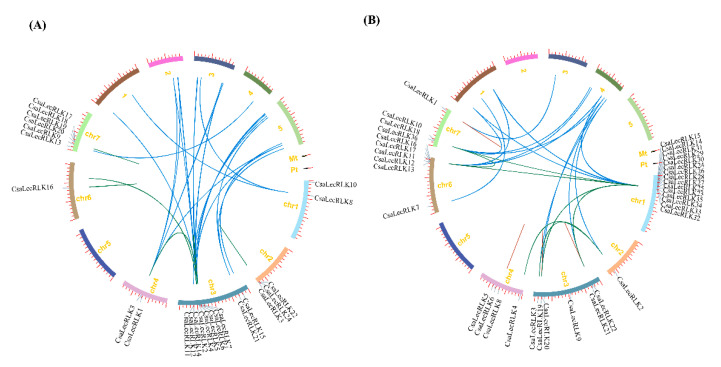
The collinear relationship of (**A**) L-type LecRLK and (**B**) G-type LecRLK proteins between cucumber and *Arabidopsis* (indicated with blue lines), and within cucumber (indicated with green color). The red line indicates tandem duplication between G-type LecRLK proteins of cucumber.

**Figure 4 ijms-22-06585-f004:**
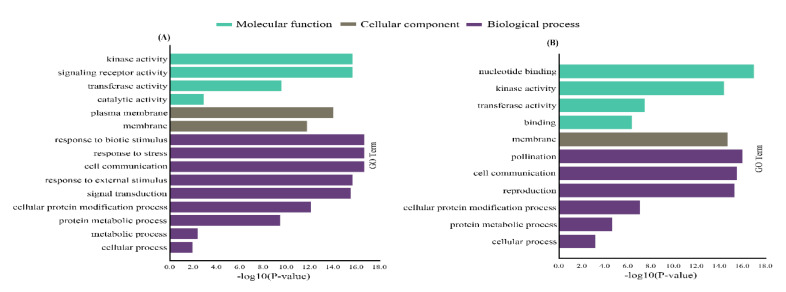
Gene ontology-based enrichment analysis of (**A**) L-type LecRLK and (**B**) G-type LecRLK proteins in cucumber. The proteins are functionally characterized into molecular function (green bars), cellular component (grey bar), and biological process (purple bars).

**Figure 5 ijms-22-06585-f005:**
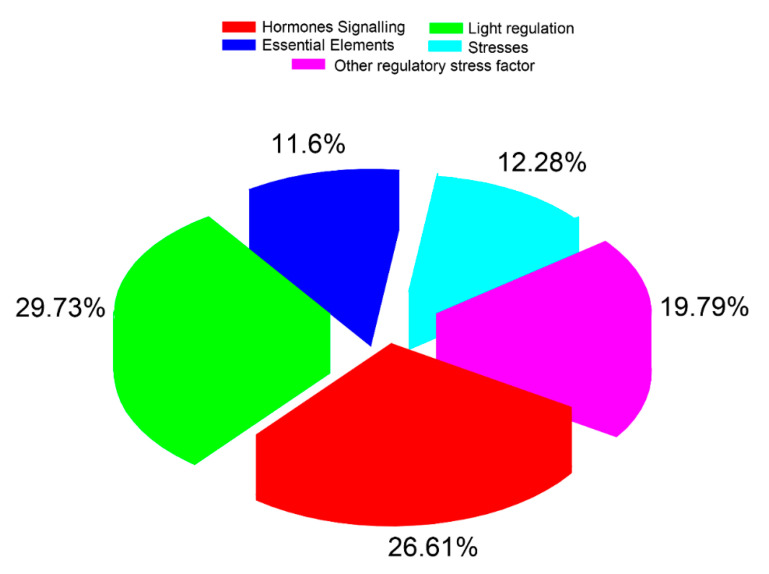
Various cis elements found in cucumber using the PlantCARE database.

**Figure 6 ijms-22-06585-f006:**
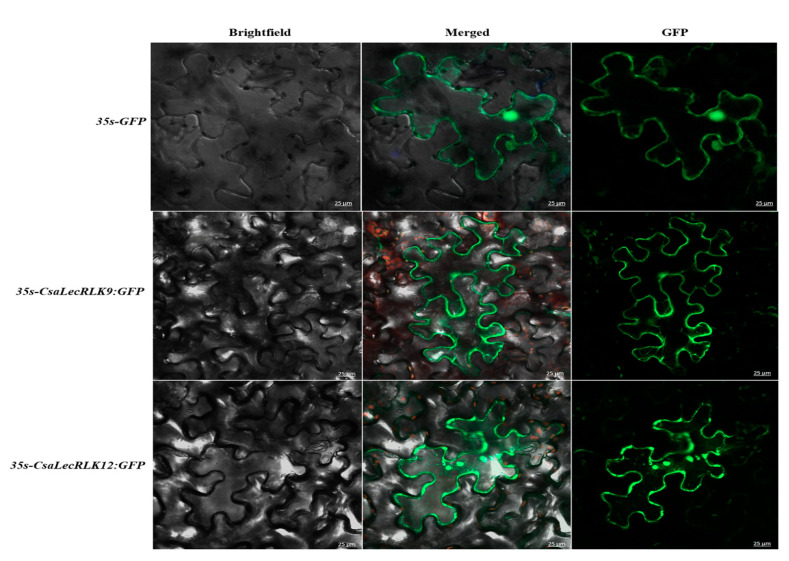
Subcellular localization of LecRLKs in the plasma membrane of tobacco (*N. benthamiana*) leaves. The green fluorescent proteins (GFP), both alone and along with CsaLecRLK9 and CsaLecRLK12, were transiently co-expressed in tobacco. A confocal microscope was used to take the picture with a bar scale of 25 µm.

**Figure 7 ijms-22-06585-f007:**
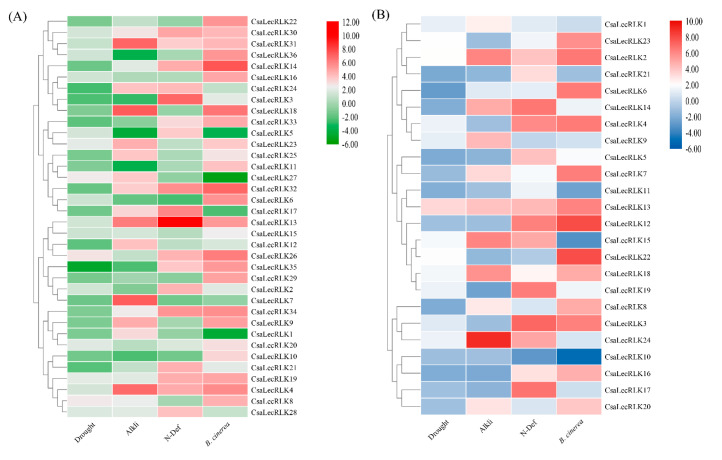
RNA-sequencing data on biotic (*Botrytis cinerea*), and abiotic (drought, alkali, and nitrogen deficiency) stresses for both G-type (**A**) and L-type (**B**) CsaLecRLKs. The heatmap was generated on the Log2 of RPKM values using Rstudio (A package of R, USA).

**Figure 8 ijms-22-06585-f008:**
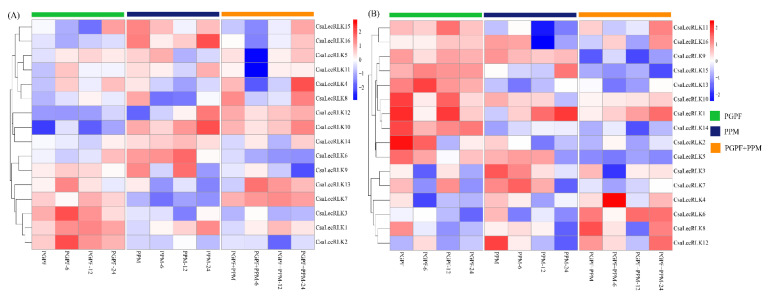
qRT-PCR validation of (**A**) L-type LecRLK and (**B**) G-type LecRLK genes in cucumber in response to *T. harzianum* (PGPF), *Golovinomyces orontii* (PPM), and a combination of the two (PGPF+PPM) treatments, respectively. The heatmap was generated based on the Log2 value using the Rstudio program (A package of R, USA).

**Table 1 ijms-22-06585-t001:** Duplications of the *LecRLK* genes in *Cucumis sativus*.

Gene 1	Gene 2	*Ks*	*Ka*	*Ka/Ks*	Divergence Time	Selection Pressure	Gene Duplications
**Between L-type genes**							
*CsaLecRLK1*	*CsaLecRLK5*	1.19	0.40	0.33	39.67	Purifying Selection	WGD or Segmental
*CsaLecRLK14*	*CsaLecRLK16*	0.91	0.53	0.58	30.33	Purifying Selection	WGD or Segmental
*CsaLecRLK2*	*CsaLecRLK4*	0.51	0.55	1.09	17.00	Positive Selection	Tandem
*CsaLecRLK6*	*CsaLecRLK7*	0.34	0.38	1.10	11.33	Positive Selection	Tandem
*CsaLecRLK18*	*CsaLecRLK19*	0.18	0.28	1.55	6.00	Positive Selection	Tandem
*CsaLecRLK22*	*CsaLecRLK23*	0.52	0.14	0.26	17.33	Purifying Selection	Tandem
*CsaLecRLK8*	*CsaLecRLK9*	0.79	0.38	0.48	26.33	Purifying Selection	Dispersed
*CsaLecRLK10*	*CsaLecRLK13*	0.84	0.55	0.65	28.00	Purifying Selection	Dispersed
*CsaLecRLK20*	*CsaLecRLK21*	1.73	0.30	0.17	57.67	Purifying Selection	Dispersed
*CsaLecRLK3*	*CsaLecRLK11*	0.89	0.59	0.67	29.67	Purifying Selection	Proximal
*CsaLecRLK12*	*CsaLecRLK17*	0.85	0.61	0.72	28.33	Purifying Selection	Proximal
**Between G-type genes**							
*CsaLecRLK2*	*CsaLecRLK3*	0.22	0.41	1.85	7.33	Positive Selection	WGD or Segmental
*CsaLecRLK4*	*CsaLecRLK10*	0.77	0.65	0.84	25.67	Purifying Selection	WGD or Segmental
*CsaLecRLK12*	*CsaLecRLK15*	0.97	0.45	0.46	32.33	Purifying Selection	WGD or Segmental
*CsaLecRLK17*	*CsaLecRLK18*	1.36	0.43	0.31	45.33	Purifying Selection	WGD or Segmental
*CsaLecRLK19*	*CsaLecRLK20*	0.98	0.16	0.16	32.67	Purifying Selection	WGD or Segmental
*CsaLecRLK22*	*CsaLecRLK25*	0.65	0.49	0.76	21.67	Purifying Selection	WGD or Segmental
*CsaLecRLK27*	*CsaLecRLK33*	0.81	0.49	0.60	27.00	Purifying Selection	WGD or Segmental
*CsaLecRLK35*	*CsaLecRLK36*	1.03	0.47	0.46	34.33	Purifying Selection	WGD or Segmental
*CsaLecRLK8*	*CsaLecRLK11*	0.81	0.58	0.72	27.00	Purifying Selection	Tandem
*CsaLecRLK13*	*CsaLecRLK16*	0.67	0.53	0.79	22.33	Purifying Selection	Tandem
*CsaLecRLK21*	*CsaLecRLK23*	1.07	0.48	0.44	35.67	Purifying Selection	Tandem
*CsaLecRLK24*	*CsaLecRLK26*	0.41	0.38	0.93	13.67	Purifying Selection	Tandem
*CsaLecRLK28*	*CsaLecRLK29*	0.53	0.60	1.12	17.67	Positive Selection	Tandem
*CsaLecRLK30*	*CsaLecRLK31*	0.48	0.14	0.29	16.00	Purifying Selection	Tandem
*CsaLecRLK6*	*CsaLecRLK7*	0.96	0.57	0.59	32.00	Purifying Selection	Dispersed
*CsaLecRLK5*	*CsaLecRLK14*	0.54	0.61	1.13	18.00	Positive Selection	Proximal

## Data Availability

The transcriptomic data used in this study can be accessed with its SRA ID number from the NCBI database. Other Appendix A is available in the additional files.

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
