# Peer review of "Genome-Wide Identification, Diversification, and Expression Analysis of Lectin Receptor-Like Kinase (LecRLK) Gene Family in Cucumber under Biotic Stress"

_ijms, 2021, doi:10.3390/ijms22126585_

Round 1

Reviewer 1 Report

Microorganisms interact with the plant and form a dynamic system in which, e.g. processes of direct pathogen growth limitation as well as processes exerting indirect effect on the growth of both organisms occur simultaneously. These processes enhance the induction of plant resistance to pathogens and regulate mutual metabolite production. Interaction between the plant and the pathogen depends on the expression level of one or several genes belonging to the so-called “pathogenicity genes”. These genes are essential for disease development and condition pathogen functioning in the plant in vivo. However, they are not important for the phytopathogen and its life cycle in vitro.

The researchers showed that some LecRLK gene may function against the biotic stress tolerance in cucumber, which could help in breeding program for developing durable powdery mildew resistant cucumber improved hybrids.

In general terms the topic of the reviewed article is interesting. The  manuscript was prepared with care and its content contains a lot of valuable information. The work does not raise any scientific or substantive reservations.

- My comments are related to the current nomenclature of fungi (according to Index Fungorum).

Currently:

No Erysiphe cichoracearum but Golovinomyces orontii (Castagne) V.P. Heluta

The paper needs some editorial corrections.

Some comments and recommendations have been included in the text.

I recommend the publication of this manuscript in the International Journal of Molecular Sciences after minor corrections.

Author Response

Responses to the Reviewers’ Comments

The authors would like to thank the Reviewer’s for his/her constructive comments and suggestions that have helped us improve our manuscript. An extensive revision has been undertaken and incorporated all the corrections and suggestions raised by the Reviewer’s in the revised manuscript.

Editor and Reviewer comments:    

Reviewer #1

Microorganisms interact with the plant and form a dynamic system in which, e.g. processes of direct pathogen growth limitation as well as processes exerting indirect effect on the growth of both organisms occur simultaneously. These processes enhance the induction of plant resistance to pathogens and regulate mutual metabolite production. Interaction between the plant and the pathogen depends on the expression level of one or several genes belonging to the so-called “pathogenicity genes”. These genes are essential for disease development and condition pathogen functioning in the plant in vivo. However, they are not important for the phytopathogen and its life cycle in vitro. The researchers showed that some LecRLK gene may function against the biotic stress tolerance in cucumber, which could help in breeding program for developing durable powdery mildew resistant cucumber improved hybrids. In general terms the topic of the reviewed article is interesting. The manuscript was prepared with care and its content contains a lot of valuable information. The work does not raise any scientific or substantive reservations.

Response: We are very glad that the Reviewer highly evaluated our manuscript, and provided constructive comments and suggestions that have helped us improve the quality of our manuscript.

Comment 1: My comments are related to the current nomenclature of fungi (according to Index Fungorum).

Currently:

No Erysiphe cichoracearum but Golovinomyces orontii (Castagne) V.P. Heluta

Response: We appreciate this Reviewer for this critical observation of our manuscript which we totally agree. As recommended by you the scientific name of the pathogen (Erysiphe cichoracearum is replaced with Golovinomyces orontii) along with the authority as well for Trichoderma harzianum at its first usage has been incorporated in the revised manuscript.

The paper needs some editorial corrections.

Comment 2: Some comments and recommendations have been included in the text.

Response: We thank the Reviewer for editing our manuscript that had helped us to improve the overall quality of the manuscript.Further, all the corrections/editing/suggestions were incorporated in the revised manuscript.

I recommend the publication of this manuscript in the International Journal of Molecular Sciences after minor corrections.

Response: Many thanks for positively evaluating our manuscript.

Reviewer 2 Report

The study authored by Muhammad Salman Haider et al., investigates the genome-wide survey of LecRLK gene family in Cucumber under biotic stress and characterised L-type and G-type genes based on domain composition and diversity analysis. The title is appropriate and easily connected to this research article. The results were presented clearly. However, few corrections and clarifications are necessary to carry out for the improvement of the article.

Corrections:
1.The gene lables not clearly visible in the figures (Figure 1, 2, 3, and 8).
2.Provide the supplementary tables as Excel format. The supplementary tables are not clear in word file.
3.Additionally, provide the radial/dendogram view of phylogenetic tree (figure 1) in the supplementary file. It is difficult to see the diversity and the deviation of genes on circular tree view. modify the lable fonts.
4. Figure 4, the authors have not identified any binding proteins on molecular function of L type LecRLK?? discuss a sentence.
5.Results 2.1. title is misleading. Gene collection or retrieval and sequence composition (PROTPARAM) analysis of CsaLecRLK genes.
6.Figure 7 legend, Change "RNA-Deq data" to RNA-Seq data.
7.How the gene ontology enrichment was predicted, mention in method part.

Major concern:

1.The authors have provided brief introduction about the importance of genes on the immunity responses in other plants like Arabidopsis but not about the previous studies on cucumber plant or about the LRR-RLK genes on cucumber?
Try to include a sentence about the importance of the study focusing on PGPF and PPM diesease in the introduction.

2. Make figure lables are clear in Figure 8 and discus the results in discussion part and indicate.

Provide the sequence dataset (used in phylogenetic tree) in supplementary file.

Author Response

Responses to the Reviewers’ Comments

The authors would like to thank the Reviewer’s for his/her constructive comments and suggestions that have helped us improve our manuscript. An extensive revision has been undertaken and incorporated all the corrections and suggestions raised by the Reviewer’s in the revised manuscript.

Reviewer #2

The study authored by Muhammad Salman Haider et al., investigates the genome-wide survey of LecRLK gene family in Cucumber under biotic stress and characterised L-type and G-type genes based on domain composition and diversity analysis. The title is appropriate and easily connected to this research article. The results were presented clearly. However, few corrections and clarifications are necessary to carry out for the improvement of the article.

Response: We would like to express our special thanks to the Reviewer for evaluating our manuscript positively. Also thanks for providing constructive comments that have helped us improve the quality of our manuscript. We have taken all our efforts to revise the manuscript, taking into account all the comments and suggestions of the Reviewer.

Corrections:
Comment 1:  The gene lables not clearly visible in the figures (Figure 1, 2, 3, and 8).
Response: Thank you very much for this comment and observation. With due respect to the Reviewer we have now modified/ replaced with the new figures with good visibility of the text/labels.

Comment 2: Provide the supplementary tables as Excel format. The supplementary tables are not clear in word file.
Response: Thank you very much for this critical observation. As per your suggestion, the supplementary data has now been supplied in the Excel file format.  

Comment 3: Additionally, provide the radial/dendogram view of phylogenetic tree (figure 1) in the supplementary file. It is difficult to see the diversity and the deviation of genes on circular tree view. modify the lable fonts.
Response: Thank you very much for this query and suggestion. Following the Reviwer query, we have included the phylogenetic tree with dendrogram view as a supplementary figure for both L-type and G-type LecRLKs. We have also enlarged the font size of circular phylogenetic tree for clear visibility.

Comment 4: Figure 4, the authors have not identified any binding proteins on molecular function of L type LecRLK?? discuss a sentence.
Response: Thank you very much for this comment. Figure 4 in the manuscript represents all three main GO terms (e.g., cellular components, biological process and molecular function) along with their binding proteins. The same has been discussed in the revised manuscript of section 2.4.

Comment 5: Results 2.1. title is misleading. Gene collection or retrieval and sequence composition (PROTPARAM) analysis of CsaLecRLK genes.
Response: Very sorry to mention that we do not state the above subtitle in our manuscript. The title of the section 2.1. refers to “2.1. Identification and structural analysis of CsaLecRLK genes”.

Comment 6: Figure 7 legend, Change "RNA-Deq data" to RNA-Seq data.
Response: We are very sorry for this topographical error. According to your suggestion, correction has been made in the revised manuscript.

Comment 7: How the gene ontology enrichment was predicted, mention in method part.

Response: Thank you very much for this comment and observation. We have now incorporated the GO enrichment prediction analysis in M&M section of the revised manuscript.

Major concern:

Comment 8: The authors have provided brief introduction about the importance of genes on the immunity responses in other plants like Arabidopsis but not about the previous studies on cucumber plant or about the LRR-RLK genes on cucumber?
Try to include a sentence about the importance of the study focusing on PGPF and PPM disease in the introduction.

Response: We again thank the Reviewer for his/her critical observation. All the relevant information has now been added in the introduction part (line77-85). However, there is no information available on the overexpression cucumber LecRLK against biotic stresses.

In addition, the importance of the study focusing on the beneficial characters of PGPF with that of pathogen or disease (PPM) causing problem on host plant has been incorporated in the introduction section of revised manuscript.

Comment 9: Make figure lables are clear in Figure 8 and discus the results in discussion part and indicate.

Response: We highly appreciate the Reviewer for this suggestion. As per your suggestion, we have modified all the figures and discussion part of figure 8 in the revised manuscript.

Comment 10: Provide the sequence dataset (used in phylogenetic tree) in supplementary file.

Response: Thank you again for this observation. Following your recommendation, we have included both L-type and G-type sequence dataset for cucumber in the supplementary Table S4.

Round 2

Reviewer 2 Report

Thanks and appreciation to the authors have performed all the correction and changes according to the comments. Again, few corrections and modification are needed for the article.

Results 2.1. Title is misleading. Gene collection or retrieval and sequence composition (PROTPARAM) analysis of CsaLecRLK genes.

Response: Very sorry to mention that we do not state the above subtitle in our manuscript. The title of the section 2.1. refers to “2.1. Identification and structural analysis of CsaLecRLK genes”.

Comments regarding the authors response:

The authors were explaining the sequence analysis in the Results section 2.1, by providing the justification of PROTPARAM results. The authors have submitted the sequence to predict the sequence composition like molecular weight (MW; kDa), isoelectric point (pI), and grand average of hydropathocity 117 (GRAVY) using PROTPARAM server. Which is not correct to state the title as structural analysis. So, the authors need to clarify, where is the structural part (about the protein structure) was presented in the paragraph (line 110-130) in the revised manuscript? please.

It will be interesting to see the evolutionary deviation between L-type LecRLK (24) and G-type LecRLK (36) proteins in cucumber. So, the authors try to submit the phylogenetic tree (circular or dendo) in the supplementary section and discuss a sentence in discussion part.

Author Response

Reviewer#2

Comments regarding the authors response:

Comment 1: The authors were explaining the sequence analysis in the Results section 2.1, by providing the justification of PROTPARAM results. The authors have submitted the sequence to predict the sequence composition like molecular weight (MW; kDa), isoelectric point (pI), and grand average of hydropathocity 117 (GRAVY) using PROTPARAM server. Which is not correct to state the title as structural analysis. So, the authors need to clarify, where is the structural part (about the protein structure) was presented in the paragraph (line 110-130) in the revised manuscript? please.

Response: We are extremely sorry for misunderstood your query during first revision. We highly appreciate this Reviewer for bringing this error into our notice with which we totally agree. Since, we have NOT performed the structure analysis of the LecRLK proteins, we have modified the title as “2.1. Identification and physicochemical properties of CsaLecRLK genes

Comment 2: It will be interesting to see the evolutionary deviation between L-type LecRLK (24) and G-type LecRLK (36) proteins in cucumber. So, the authors try to submit the phylogenetic tree (circular or dendo) in the supplementary section and discuss a sentence in discussion part.

Response: Thank you very much for this critical comment and observation. With due respect to the Reviewer, we have incorporated the phylogenetic tree in the supplementary section and following sentence regarding evolution and family expansion of LecRLKs in angiospermsin the Discussion section of the revised manuscript:

“In cucumber, G-type LecRLKs (36) proteins are over L-type LecRLks (24) which is different in Arabidopsis (L-type: 42 vs G-type: 32). Hence, the number of LecRLK proteins in angiosperm is variable, which could be due to difference in selection pressure and expansion rate of the genome. The expansion between G-type and L-type greatly varies and ranges from 0.085-0.323% for L-type and 0.117-0.449% for G-type, indicating that G-type LecRLKs expanded to a significant scale as compared to L-type LecRLks. However, the expansion of LecRLK gene family in highly uncoordinated in rice and Arabidopsis because orthologous gene pairs of this family in both species expanded at variable rates [4]”.  

Round 3

Reviewer 2 Report

The authors were provided all the responses and correction in the revised manuscript.

However, the phylogenetic tree of L-type LecRLK proteins and G-type LecRLK proteins were not included in the supplementary files.

The Ks and Ka values are needed to separate in the Table 1 (revised article).

Figure 8 is not visible in the revised article.

Furthermore, line 476 supplementary materials: Figures S1-S4 change to Figures S1-S6.

Author Response

Reviewer Comments:

Comment 1: However, the phylogenetic tree of L-type LecRLK proteins and G-type LecRLK proteins were not included in the supplementary files.

Response: Thank you so much for this enquiry. We have already provided these two figures in the Supplementary files. Anyways, we have again re-submitting the supplementary figures S1 (L-type LecRLK) and S2 (G-type LecRLK) which represents the phylogenetic trees and their protein sequences have been provided in Table S4. Since, the size of the Figure S1 and S2 are large, we are also providing PDF files of the figures separately in the Supplementary folder for easy understanding.

Comment 2: The Ks and Ka values are needed to separate in the Table 1 (revised article).

Response: We thank the Reviewer for this suggestion. To meet the Reviewers suggestion, we have now modified table 1 in the revised manuscript for easy understanding.

Comment 3: Figure 8 is not visible in the revised article. 

Response: All the figures in the manuscript are supplied with high resolution of 600 dpi. Since, the space provided in the manuscript is limited, we have now replaced the Figure 8 by increasing its size in the revised manuscript.

Comment 4: Furthermore, line 476 supplementary materials: Figures S1-S4 change to Figures S1-S6.

Response: We appreciate the Reviewer for his/her critical observation of our manuscript. We have carefully checked, and renamed the supplementary figures/tables and their numbers in order (Figure S1-S5, Table S1-S5) in the revised manuscript.
